# High-severity wildfires in temperate Australian forests have increased in extent and aggregation in recent decades

**Bang Nguyen Tran** [1,2]*, **Mihai A. Tanase**[1,3], **Lauren T. Bennett**[4], **Cristina Aponte**[1,5]

**1** School of Ecosystem and Forest Sciences, University of Melbourne, Richmond, Victoria, Australia,
**2** Faculty of Environment, Vietnam National University of Agriculture, Trauquy, Gialam, Hanoi, Vietnam,
**3** Department of Geology, Geography and Environment, University of Alcala, Alcala de Henares, Spain,
**4** School of Ecosystem and Forest Sciences, The University of Melbourne, Creswick, Victoria, Australia,
**5** National Institute for Research and Development in Forestry "Marin Dracea", Voluntari, Ilfov, Romania

* ntran6@student.unimelb.edu.au

**Data Availability Statement:** All relevant data are within the manuscript and its Supporting Information files

## Abstract

Wildfires have increased in size and frequency in recent decades in many biomes, but have they also become more severe? This question remains under-examined despite fire severity being a critical aspect of fire regimes that indicates fire impacts on ecosystem attributes and associated post-fire recovery. We conducted a retrospective analysis of wildfires larger than 1000 ha in south-eastern Australia to examine the extent and spatial pattern of high-severity burned areas between 1987 and 2017. High-severity maps were generated from Landsat remote sensing imagery. Total and proportional high-severity burned area increased through time. The number of high-severity patches per year remained unchanged but variability in patch size increased, and patches became more aggregated and more irregular in shape. Our results confirm that wildfires in southern Australia have become more severe. This shift in fire regime may have critical consequences for ecosystem dynamics, as fire-adapted temperate forests are more likely to be burned at high severities relative to historical ranges, a trend that seems set to continue under projections of a hotter, drier climate in south-eastern Australia.

## Introduction

Wildfire shapes landscape patterns and ecosystem processes as it determines both vegetation distribution and structure [1, 2]. Changes in wildfire activity may alter mortality and regeneration patterns, initiating new successional pathways that ultimately lead to shifts in vegetation composition and landscape attributes [3]. Many studies over the past decades have reported a change in wildfire activity including increases in the frequency, size, and duration of wildfires, as well as the length of the fire season [4–8]. Such increases have been linked to climate change, which influences key fire drivers like fuel accumulation and availability [9–11]. Models based on climate change projections suggest that this trend in increasing fire activity will continue into the future [3, 12–15] posing threats to forest resilience, including shifts to lower density forests or non-forest states [16–18].

**Funding:** The authors would like to acknowledge the financial support of the Melbourne Research Scholarship program, the Vietnam International Education Cooperation Department (VIED) scholarship, and the Integrated Forest Ecosystem Research program, supported by the Victorian Department of Environment, Land, Water and Planning. The funders had no role in study design, data collection and analysis, decision to publish, or preparation of the manuscript.

**Competing interests:** The authors have declared that no competing interests exist

Fire severity is a wildfire attribute that quantifies the degree of environmental change caused by fire including immediate fuel consumption and carbon emissions and longer-term impacts on vegetation mortality, successional pathways, and soil substrate [19]. Wildfire severity is spatially heterogeneous and can range from partial litter consumption and light scorching of understorey vegetation to near complete mortality of canopy trees [19–21]. Fire severity and the spatial configuration of severity classes have critical implications for fire-related resilience and potential degradation of ecosystems [21–25]. Wildfire severity is related to fire intensity, which is driven by fuel, climate, and weather [26–29]. As such, fire severity, as for other components of fire regimes, has likely been affected by changing climates in recent decades [30, 31]. In contrast to the large number of studies that have documented recent increases in wildfire area and frequency [9, 32–34], comparatively fewer studies, mostly focused on North America forests, have investigated trends in fire severity, some indicating increases while others indicating no change or decreases [35–37]. Changes in wildfire severity can influence ecological processes by affecting the trajectory of postfire vegetation succession, leading to reductions in forest cover and even conversions to non-forested vegetation [38, 39]. A better understanding of changes in fire severity is crucial to foresee the future pathways of forest systems [40–44].

Australia is one of the most fire-prone countries worldwide [45, 46] with 30.4 million hectares burned across Australia in 2019–2020 alone [47]. Studies have highlighted how climate change has and will continue to impact Australian fire weather and fire activity [31, 48, 49] with fires predicted to become larger and more frequent [50–52]. Whether fires have also become more severe remains largely undocumented. This study's principal objective was to examine patterns in high-severity fires in temperate forests of the state of Victoria, south-eastern Australia over the last three decades. Specifically, we addressed three questions: 1) Has the area burnt by high- severity fire in temperate forests of Victoria increased in the last 30 years?; 2) Has the spatial configuration of high-severity patches in the landscape changed in the last 30 years; and 3) Are the observed trends consistent across bioclimatic regions?

## Materials and methods

### Study area and forest types

This study was conducted across the state of Victoria, south-eastern Australia, an area that encompasses 237,659 km$^2$, ranges from 0 to 1986 m a.s.l in elevation and comprises several geographical bioregions with differing geology, soils, climate, and predominant vegetation (Table 1 and Fig 1) [53]. Climate across Victoria is temperate with warm to hot summers (average maximum temperature between 16˚C and 30˚C; [54]). The annual mean temperature ranges from 12.6˚C in the south-east region to 14.7˚C in the north and north-west regions of the state [55]. The mean annual precipitation varies from 500 to 2,200 mm, with precipitation over 1000 mm in the mountainous areas of the Great Dividing Range [56]. Over the past few decades, Victoria has become warmer and drier, consistent with global trends, and these trends are likely to continue [57–59].

Vegetation affected by the studied wildfires was predominantly comprised of a range of *Eucalyptus* forests of varying composition, structure and post-fire regeneration strategies [60] (Table 1). These included Mallee, with low canopy height (7 m) and sparse canopy cover (25%), Woodlands with medium canopy height (15 m) and sparse canopy cover, Open forests, with medium to tall canopy height (10–30 m) and mid-dense canopy cover (30–70%) and Closed forests, with tall canopy height (30 m) and dense canopy cover (70–100%) [61]. Obligate seeder tree species are dominant in Closed forest whereas resprouter eucalypts (basal or epicormic) are dominant in all other forest types [60, 62, 63].

**Table 1. Characteristics of the bioregions in the study area affected by the selected 162 fires.**

| | Bioregion | Major forest types [a] | Height (m) | Projective Foliage Cover (%) | Regeneration strategy [b] | Elevation (m) | MAT (°C) | MAP (mm) | No of fires | Total burnt area (ha) | Total high-severity burnt area (ha) |
|---|---|---|---|---|---|---|---|---|---|---|---|
| AA | Australian Alps | High Altitude Shrubland/ Woodland | 15 | 10–30 | R | 844–1996 | 4.5–12.6 | 712–1996 | 9 | 1,426,791 | 290,073 |
| | | Riverine Woodland/Forest | 15 | 10–30 | R | | | | | | |
| MDD | Murray Darling Depression | Lowan Mallee | 7 | 10–30 | R | 265–690 | 12.8–17.2 | 265–702 | 52 | 514,689 | 358,238 |
| | | Riverine Woodland/Forest | 15 | 10–30 | R | | | | | | |
| SCP | South East Coastal Plain | Riverine Woodland/Forest | 15 | 10–30 | R | 492–1260 | 11.4–14.9 | 494–1306 | 10 | 40,375 | 8,745 |
| SEC | South East Corner | Moist Forest | 30 | 70–100 | S | 664–1184 | 7.3–15.2 | 656–1292 | 17 | 170,045 | 18,700 |
| | | Riverine Woodland/Forest | 15 | 10–30 | R | | | | | | |
| SEH | South Eastern Highlands | Grassy/Heathy Dry Forest | 10–30 | 10–30 | R | 681–1922 | 6.6–14.8 | 645–1942 | 17 | 995,133 | 170,452 |
| | | Moist Forest | 30 | 70–100 | S | | | | | | |
| VM | Victorian Midlands | Forby Forest | 15–30 | 30–70 | R | 418–1411 | 8.5–15.3 | 418–1490 | 46 | 404,363 | 156,083 |
| VVP | Victorian Volcanic Plain | Moist Forest | 30 | 70–100 | S | 477–1026 | 11–14.9 | 476–1026 | 11 | 165,003 | 79,022 |

Bioregion name and acronym [53], major forest types in each bioregion affected by the selected wildfires, height, projective foliage cover and regeneration strategy of the dominant species in each forest type, elevation range, mean annual temperature (MAT) and annual precipitation (MAP) range [65]; Number of wildfires included in this study (i.e. 162 wildfires greater than 1000 ha, occurred between 1987 and 2017 and with available Landsat imagery) and their cumulative total [64] and high-severity burnt area (as estimated in this study).

[a] Major forest types were adopted from EVD names and associated structural data [66]. Dominant tree species were derived from the Ecological Vegetation Classes (EVC) benchmarks database [67];

[b] R: resprouter; S: obligate seeder, classifications based on predominant fire-response traits of dominant tree species [62, 68, 69].

### Fire history dataset

We used the wildfire history data available from the Victorian Department of Environment, Land, Water & Planning ('DELWP'; [64]). Data contained the spatial extent of wildfires since 1926 and, for the most recent fires (from 1998 onward), the start date of the fire. For this study we selected the subset of wildfires that occurred between 1987 and 2017 and that had a minimum burned area of 1000 ha to ensure the fire size was sufficient to include multiple fire-severity levels. That amounted to 211 wildfires that were used to assess changes in the number of fires per year and mean fire size between 1987 and 2017. Each fire was classified according to its dominant bioregion [53]. For the purpose of assessing changes in fire severity, 32 of the 211 wildfires were discarded because pre- or post-fire remote sensing images were unavailable, and 11 were discarded because clouds covered more than 25% of the fire affected area, which may affect the spatial metrics assessed in our study. In total, a subset of 162 wildfires, with at least two fires per year over the past three decades, was used to generate fire-severity maps and analyse changes in severity patterns.

### Remote sensing dataset and spectral indices

Wildfire severity of the selected 162 fires was mapped using Landsat TM, ETM+ and Landsat 8 imagery (30 m spatial resolution, all from Landsat Collection 1, Tier 1). Pre- and post-fire

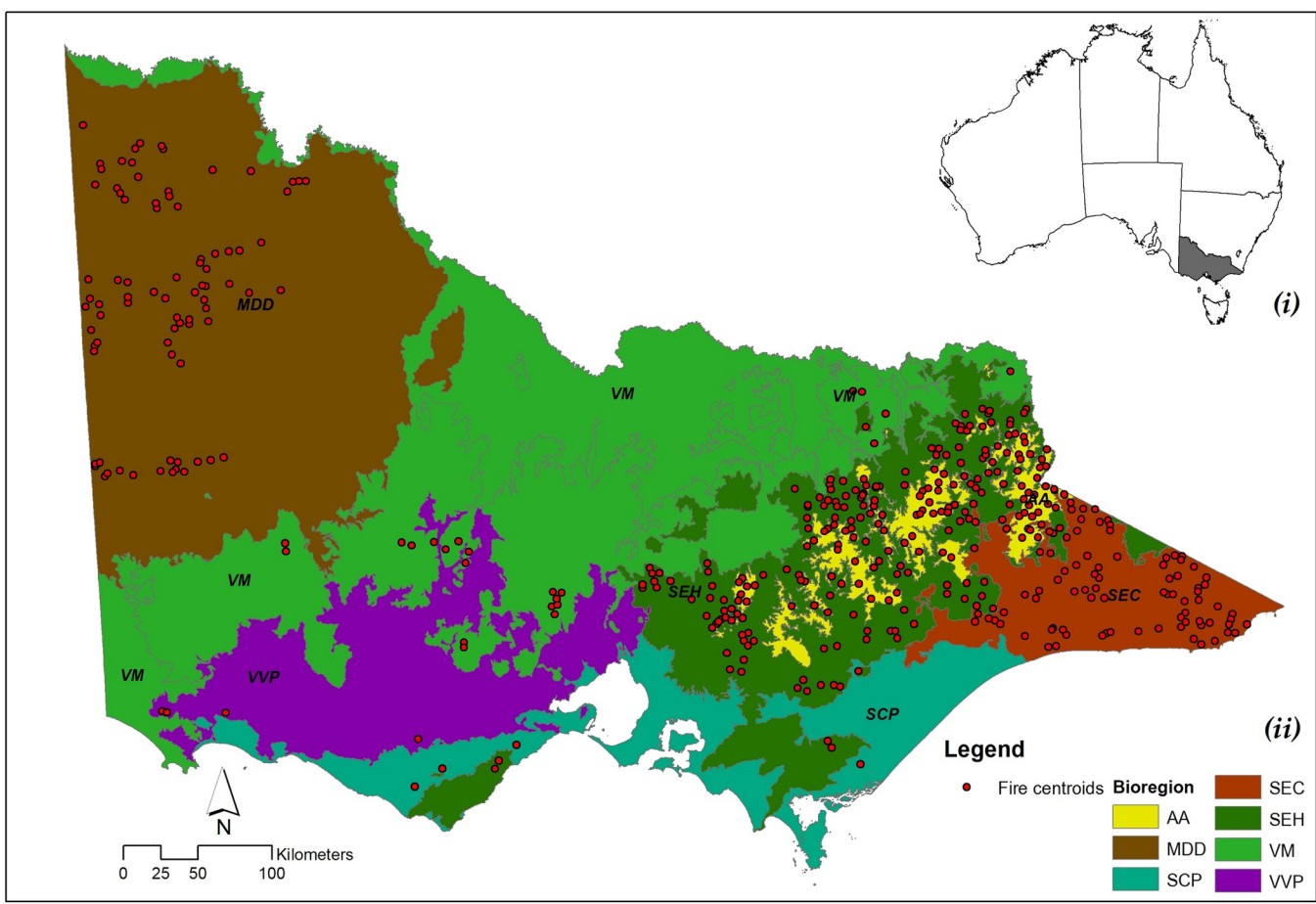

**Fig 1. Map of study area.** (**i**) Victoria highlighted (grey) in the map of Australia; (**ii**) Locations of study areas within the state of Victoria in south-eastern Australia. Red points rrepresent the centroids of the 162 wildfires investigated in this study. Colours relate to bioregions (Acronyms are defined in Table 1).

images were selected for each wildfire based on the recorded fire start dates, which were predominantly in the summer months (December to February). Images were selected within two months before and after the fire to minimise differences in forest phenology and general atmospheric conditions at the time of acquisition. When only the fire year but not start date was recorded (~13% of the fires), we conducted a visual inspection of all images available for the fire season, identified the image where the fire scar was first visible and selected that image and the previous one as post- and pre-fire images respectively for that event. A total of 347 Landsat images including 228 scenes of Landsat 5 (TM), 36 scenes of Landsat 7 (ETM+), and 83 scenes of Landsat 8 (OLI/TIRS) were selected and obtained through the US Geological Survey (USGS) EarthExplorer at http://earthexplorer.usgs.gov as higher level surface reflectance products for each fire. The images were masked for clouds and shadows using the Fmask algorithm [70], which has an accuracy of about 96% [71].

Four spectral indices, namely NBR, NDVI, NDWI, and MSAVI, and their temporal differences (i.e. delta versions, which calculate the change between pre-fire and post-fire spectral index values) were computed for each of the 162 wildfires. These indices are commonly used to assess fire severity [72–76] and were identified by the authors, in a previous study, as the optimal spectral indices for mapping fire severity in the forest types of the study area [77].

## Fire severity mapping

Severity of the wildfires in Victoria has not been consistently recorded, with historic fire severity mapping only available for nine years in the period between 1998 and 2014 [78]. To generate fire severity maps for the 162 selected wildfires ensuring the consistency of the classification we used a Random Forest model based on spectral indices that had been previously trained and validated by the authors for the same study area [61]. The reference fire-severity dataset used for training and validation was comprised of 3730 plots from eight large wildfires (>5,000 ha) that occurred between 1998 and 2009 and covered 13 forest types differing in species composition, canopy cover, canopy height and regeneration strategy. These forest types match those affected by the 162 wildfires of this study. Fire severity of the 3730 reference plots had been assessed *in situ* or visually interpreted on very high resolution orthophotos by the Department of Environment, Water & Planning (DELWP) [78]. Severity was classified as Unburnt: less than 1% of eucalypt and non-eucalypt crowns scorched; Low severity: light scorch of 1–35% of eucalypt and non-eucalypt crowns; Moderate severity: 30–65% of eucalypt and non-eucalypt crowns scorched; or High severity: 70–100% of eucalypt and non-eucalypt crowns burnt [79]. Overall, the reference data included a minimum of 20 plots for each forest type and fire-severity class combination. The Random Forest model was trained with 60% of the data and used 12 predictor variables, which included the four optimal SI indices (dNBR, dNDVI, dNDWI, and dMSAVI) and their pre- and post- fire values. Model accuracy was tested on the remaining 40% of the data that had been left for model validation. Accuracy for high-severity mapping was very high, with a commission error (plots wrongly attributed to high severity) of 0.06 and an omission error (high severity plots incorrectly classified) of 0.18.

## Metrics of high-severity fire

Based on the high-severity maps of each of the 162 wildfires, we calculated eight landscape metrics to characterize the extent and spatial configuration of the high-severity burned area. Extent metrics included total and proportional high-severity burned area. Spatial configuration metrics were calculated at the patch level, i.e. areas of high-severity fire surrounded by different severities within the wildfire boundary. Spatial configuration metrics included two patch size metrics (mean patch size, coefficient of variation of patch size), two fragmentation metrics (number of patches, and edge density—a measure of shape complexity) and two aggregation metrics (clumpiness and normalized landscape shape index–NLSI, S1 Table of S1 File). Edge density is the ratio between the total length (m) of the edges of the high-severity patches and the fire size (i.e. total wildfire area burnt at any severity; ha). Low edge density values represent simple shape (e.g. circular) and/or large patches, while large values indicate irregular and/or less continuous patches [80]. Clumpiness and NLSI, both unitless, quantify patch aggregation. The former is based on the likelihood of adjacent pixels belonging to the same class, whereas the later measures the deviation from the hypothetical minimum edge length of the class. Increasing levels of aggregation (i.e. increasing clumsiness and decreasing NLSI) represent more compact and simpler-shaped patches [80, 81]. These metrics describe different aspects of landscape configuration but were not completely independent and therefore should be interpreted jointly (S1 Table of S1 File). Spatial pattern metrics were obtained using the 'landscapemetric' package [82] in the R statistical software [83].

## Data analysis

Linear regression models were used to evaluate the trends in high-severity fire metrics from 1987 to 2017, with individual fires as the sampling unit. We built two groups of models, a

state-wide model (n = 162 fires) and separate bioregion models. The response variables for both groups of models were the extent or landscape configuration metrics of the high-severity burned area. Predictor variables included year and fire size (i.e. total wildfire area, ha) as fixed effects and bioregion as a random effect, which was only included in the state-wide mixed effects models. Fire size was included as covariate in all models as it can be related to burn patterns [27] and was not correlated with fire year (Pearson's r = -0.01). Data were transformed when needed to meet assumptions of normality (S1 Table of S1 File). All statistical tests were conducted in the statistical programming language R [83].

## Results

### Changes in area and proportion of high-severity fire over time

Based on the fire history dataset (n = 211), the number of wildfires per year larger than 1000 ha between 1987 and 2017 increased significantly (P = 0.012), a trend that was mostly due to an increase since 2000 (Fig 2). In contrast, we detected no significant change in total fire size (i.e. all fire severities combined) over that period.

Between 1987 and 2017 the area burnt by high-severity fire increased significantly ($P_{Year}$ <0.001) even when accounting for total fire size ($P_{Fire\ size}$ <0.001; Fig 3 and S1 Fig of S1 File). The same trend was observed for the proportion of the area burnt by high-severity fire ($P_{Year}$ <0.001; Fig 3). Estimated changes in the area and the proportion of area burnt by high-severity fire over time by bioregions were positive and significant (or marginally significant 0.05< P <0.1) in all cases (Fig 3 and S2-S3 Figs of S1 File). The studied bioregions supported quite distinct forest types, from wet, tall, and highly productive to dry, open, and less productive. This suggests that the observed increases in the area burnt by high-severity fire was ubiquitous across regions and did not depend on local environmental conditions or forest types.

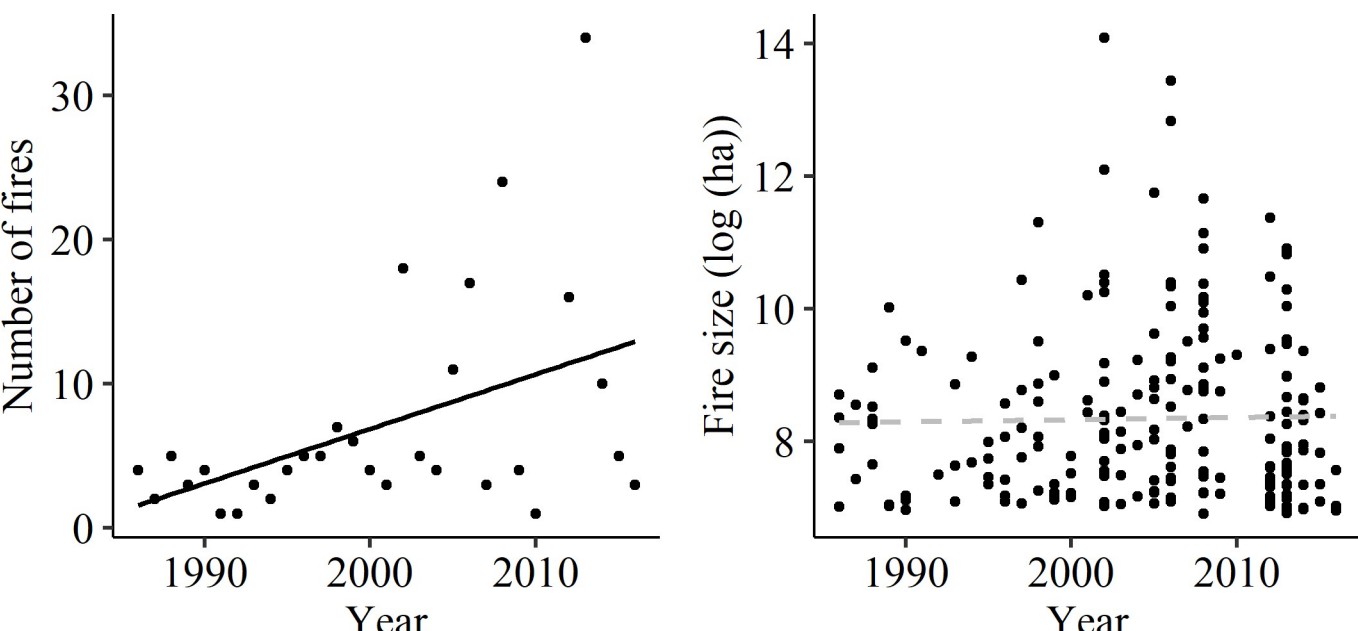

**Fig 2. Changes in the number of fires per year and fire size between 1987 and 2017.** Data includes all wildfires ≥ 1000 ha from DEWLP fire history dataset (n = 211) [64]. Solid black line indicates significant relationship (P<0.05), dashed grey line indicates no significant relationship.

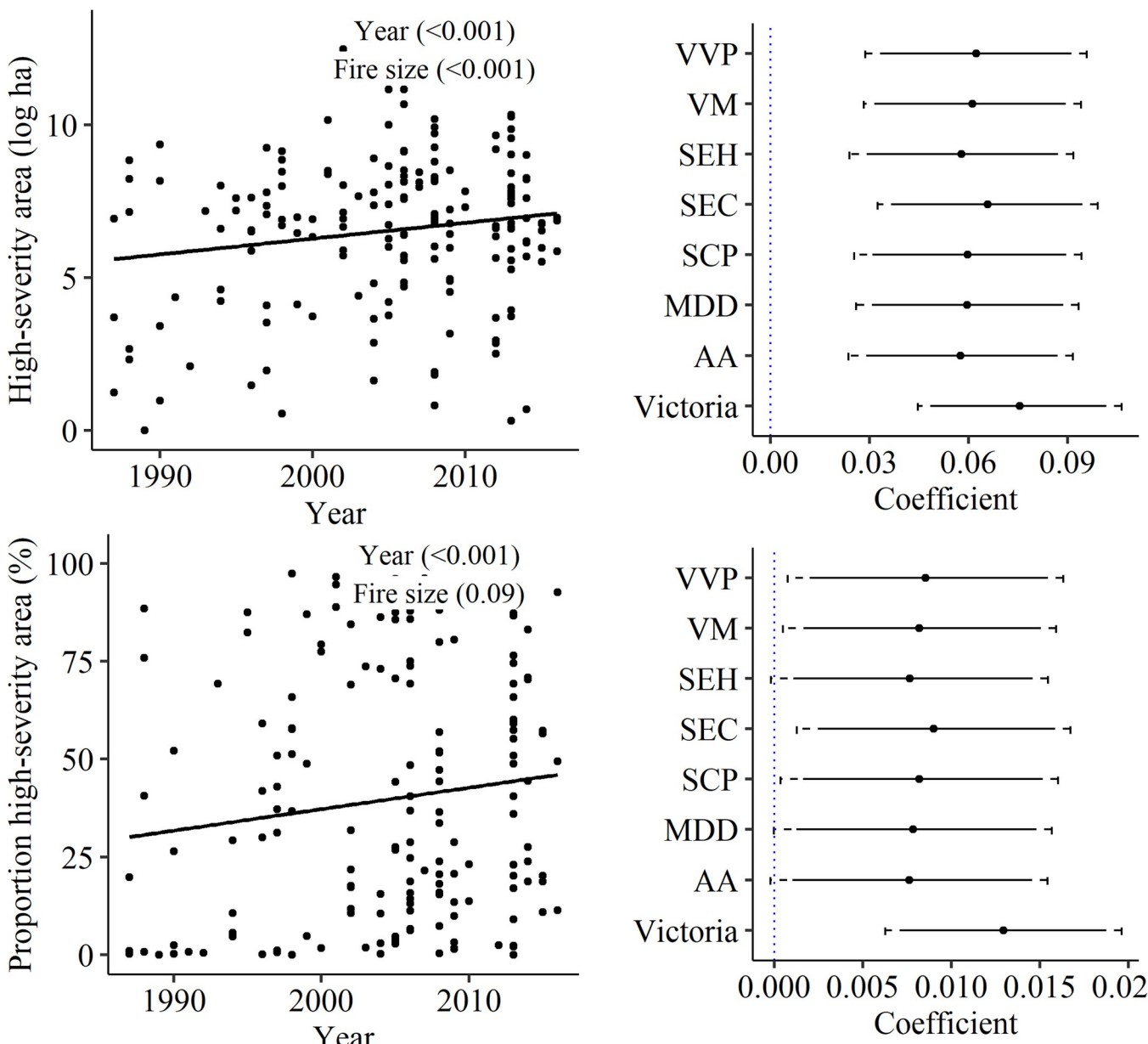

**Fig 3. Changes in the area and proportional area of high-severity fire from 1987 to 2017.** Left panels: Area and proportional area burnt by high-severity fire in each of 162 wildfires (line represents significant relationship between variables). Right panels: Standardized coefficients for high-severity area (top, log transformed) and the proportion high-severity area (bottom, arcsine transformed) indicating the relationship between area burnt and time. Each panel displays results for a single model for all regions ("Victoria") and for individual bioregions (Acronyms of bioregions are defined in Table 1); Dot points represent mean estimated coefficient along with the 90th (solid line) and 95th (dashed line) percentile intervals. Coefficients denote significant changes when interval does not include zero.

## Changes in spatial patterns of high-severity fire

We detected no changes in fragmentation of wildfires between 1987 and 2017 as evidenced by no significant increases in the number of high-severity patches, a result that was consistent across all bioregions (Figs 4 and 5 and S4 Fig of S1 File). In contrast, edge density, which is related to patch shape complexity, increased over time across Victoria ($P_{Victoria} = 0.006$), although this trend was only (marginally) significant for the SEC, VM, VVP bioregions ($0.05 < P_{Year} < 0.1$; Fig 5 and S5

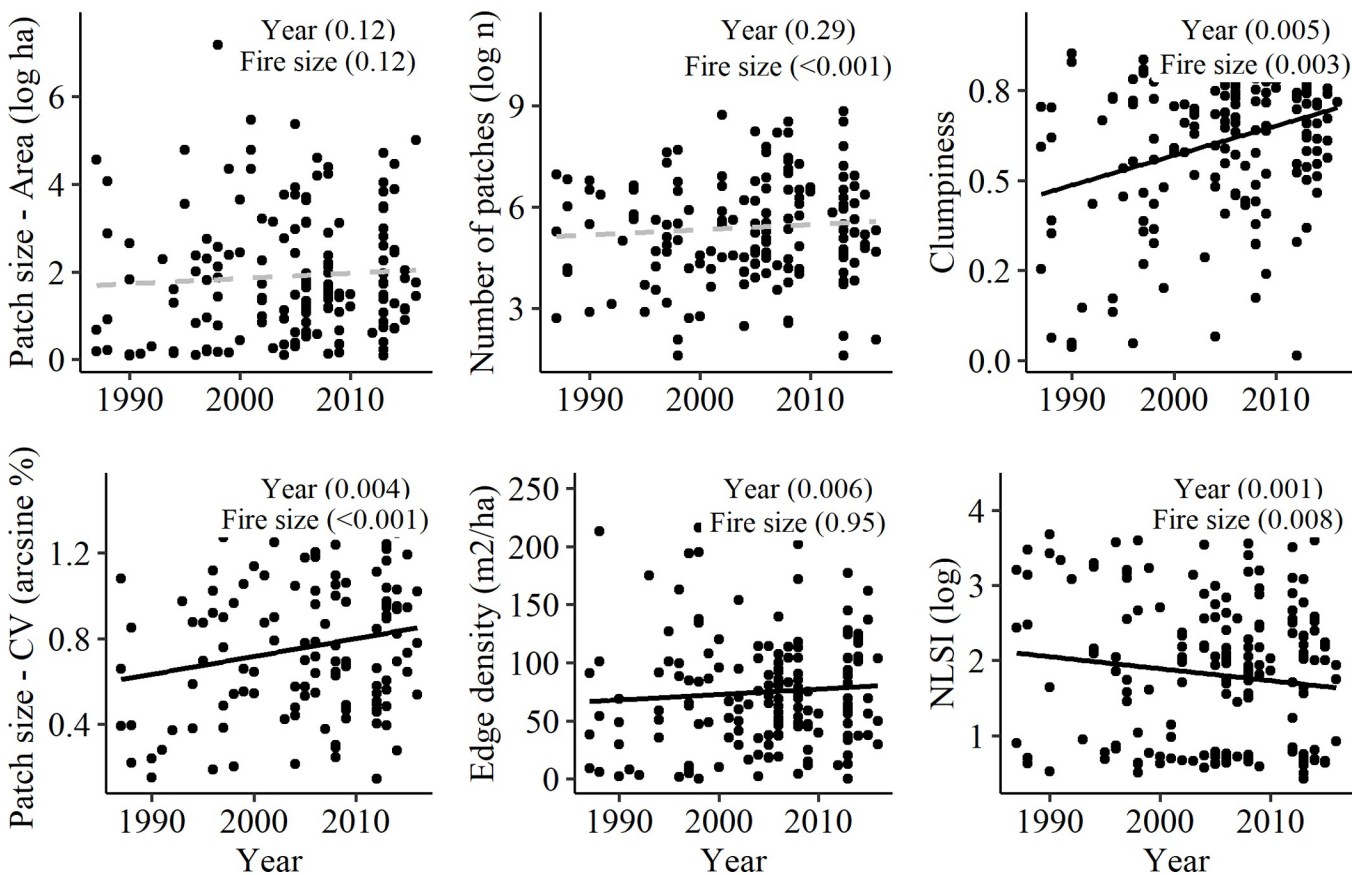

**Fig 4. Changes in high-severity spatial metrics over time.** Each subplot displays a scatterplot between the Year of the fire and the defined high-severity spatial metric. Dots represent each of the 162 wildfires. Values are the results for single mixed effects models where Year and Fire size are fixed effects and Bioregion is a random effect. Lines represent significant (solid black) or not significant (dashed grey) linear relationships.

Fig of S1 File). While mean high-severity patch size did not change significantly, the coefficient of variation of patch size, which was related to fire size, increased in all models ($P_{Year} < 0.05$ and $P_{Fire\ size} < 0.001$; Figs 4 and 5 and S6-S7 Figs of S1 File). Accordingly, we detected an increase in the size of the largest patch ($P_{Year} = 0.005$; S8 and S9 Figs of S1 File). The level of patch aggregation measured through increased clumpiness and/or decreased Normalized Landscape Shape Index (NLSI), also increased from 1987 to 2017 (Figs 4 and 5 and S10 and S11 Figs of S1 File). This trend, which was significant both at the state and bioregion level, suggests the patterns in high-severity fire changed from a more random, highly-dispersed distribution of patches towards fewer, larger patches of irregular shape that were more aggregated within the fire boundaries.

## Discussion

Our study assessed for the first-time changes in high-fire severity patterns since 1987 in Victoria, south-eastern Australia. We detected an increase in the area burnt at high-severity during that period and a shift in the landscape configuration of high-severity patches, which was consistent across most bioregions, encompassing a broad range of forest types.

### The area of high-severity fire has increased

Our results showed an increasing trend in both total and proportion of high-severity burned area between 1987 and 2017 across various temperate forests types in south-eastern Australia.

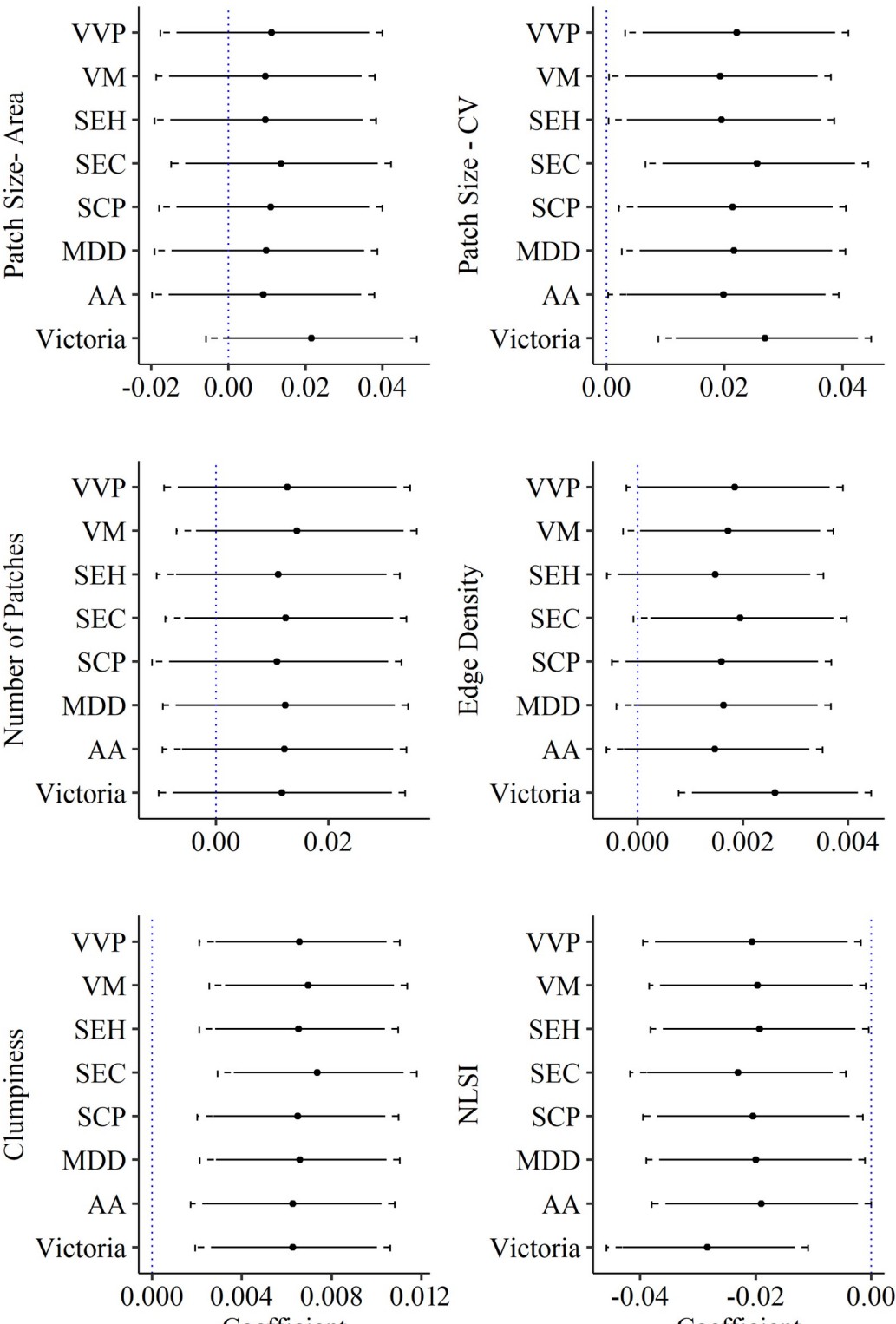

**Fig 5. Estimated coefficients for high-severity spatial metrics by bioregions.** Each panel displays results for a single model for all regions ("Victoria") and for individual bioregions (Acronyms of bioregions are defined in Table 1); Dot points represent mean estimated coefficient along with the 90th (solid line) and 95th (dashed line) percentile intervals. Coefficients denote significant changes when interval does not include zero. Spatial metrics were log transformed (Number of Patches, Mean Patch Area, Variation Patch Area, NLSI) or arcsine transformed (Edge Density).

Our findings are in contrast to similar studies conducted in the US where either an increase in fire severity was not detected [37, 84] or the detected increase was due to increasing fire size [36]. Our results also show a covariation between fire size and the extent of the area burned by high-severity fire, a pattern that has been documented before in several north American forests [4, 27, 85–87].

The increasing trends in total and proportion of high-severity burned area at the state level were consistent across all bioregions, indicating that these changes occurred irrespective of forest type and climatic region. This is in contrast to the mixed fire-severity trends assessed across regions in North America [37, 88], which have been argued to be related to fire suppression policies masking climate-change effects [84, 88].

Changes in the area of high-severity fire like those described here have been predicted to occur as a result of climate change since decades ago [89–91]. Our results confirm for the first time that wildfires in south-east Australia are indeed becoming more severe and, given projections of a hotter, drier climate [59], this pattern seems set to continue in coming decades.

## Trends in landscape configuration: Aggregation of high-severity patches

Our results showed changes in the landscape configuration of high-severity patches that were consistent at the state level and across bioregions. While we did not detect a significant shift in patch number or mean patch size, we noted an increase in patch size variability, patch shape complexity (measured as edge density) and patch aggregation (as evidenced by trends in clumpiness and NLSI). These changes suggest that the areas burned by high-severity fire have become more aggregated, more irregular in shape, and have a larger area occupied by the largest patch. Similar changes in spatial patterns of high-severity fire have also been reported in fire-severity research in North America [27, 88, 92], where increasing patch aggregation was related to the increased proportion of high-severity area [42].

## Implications of increasing high-severity fire for temperate forests in south-east Australia

Our quantified increases in high-severity burned area can lead to concerns about the resilience of Victoria's temperate forests [20, 93, 94], similar to those expressed for other forest types elsewhere [4, 92, 95]. High-severity fire influences ecosystem dynamics with effects on vegetation succession [25, 96, 97], biogeochemical processes [21, 26, 98], geomorphic processes [99, 100], and habitat availability and biodiversity [23, 101, 102]. Recent high-severity fires within our study area have led to increased mortality of fire-tolerant eucalypt trees and to an increase in the density of young trees vulnerable to subsequent fires [20, 63, 103]. If increasing trends in the extent of high-severity fire detected in our study continue, this indicates potential for large-scale changes in key structural attributes of even the most fire-tolerant forests.

High-severity fire impacts can be modulated by the size, shape, and configuration of high-severity patches. For instance, patch size and aggregation can influence runoff connectivity and post-fire sediment yields and affect the distribution of low- and moderate-severity patches that serve as refuges for fire-sensitive species [104–106]. Patch size and spatial configuration can also affect dispersal and subsequently influence vegetation succession potentially leading to forest-type conversions [107–109]. Delays in tree re-establishment following high-severity fires has been detected in non-serotinous forests of the United States and Canada due to a rapid and extensive shrub establishment via persistent soil seedbanks [109, 110]. Eucalypt forests in south-eastern Australia, including those affected by the studied wildfires, are dominated by either resprouter species that survive most fires, or obligate seeder species that rely on a canopy seedbank to regenerate after fire [63, 111]. Seed dispersal in both resprouters and obligate

seeders eucalypt forests is limited to one or two tree heights, with seeds lacking attributes to facilitate animal or wind dispersal [112]. Resprouters' seed viability decreases with fire intensity [113] and therefore regeneration in high-severity patches may depend on dispersal from adjacent moderate-severity or unburned patches (although see [20] indicating prolific regeneration from seed of resprouter eucalypts after a single high-severity wildfire). Increases in high-severity patch size though aggregation as observed in this study could hinder post-fire tree establishment by increasing distances from seed source and also altering the regeneration abiotic environment [114] contributing to feedbacks that result in an increased risk of forest-type conversion [115, 116]. Spatial configuration of high-severity patches can also influence regeneration of obligate seeder forests burnt by recurrent fires in quick succession (~20 years; [103]). In such circumstances, trees regenerating after the first fire would not have yet produced meaningful quantities of viable seed before a second fire [117], and eucalypt regeneration would rely on seed dispersal from adjacent patches. Lack of tree regeneration after short-interval fires in obligate seeder forests has been observed in the last decades with aerial sowing being required to address post-fire recovery in obligate seeder forests [118]. This highlights the impact that the observed changes in fire regimes have had on the resilience of eucalypt forests in south-eastern Australia [63, 103].

## Conclusions

Changes in high-severity fire, its extent and spatial configuration, can alter a range of ecosystem processes that interactively determine post-fire recovery, including the conversion to non-forest alternative states. Our analysis showed an increase in both the total and proportion of high-severity burned area in Victoria between 1987 and 2017. Over that period, high-severity patches have become more aggregated and more irregular in shape. These trends were consistent across bioregions encompassing a diversity of forest types. Shifts in the spatial patterns of high-severity fire over time may have cascading effects on forest ecology, highlighting the increased threat posed by changing fire regimes to forests ecosystems.

## Supporting information

**S1 File.**
(DOCX)

## Acknowledgments

We acknowledge the support of many staff and students from the School of Ecosystem and Forest Sciences at the University of Melbourne, including valuable comments and advice to improve this manuscript.

## Author Contributions

**Conceptualization:** Bang Nguyen Tran, Lauren T. Bennett, Cristina Aponte.

**Data curation:** Bang Nguyen Tran, Cristina Aponte.

**Formal analysis:** Bang Nguyen Tran.

**Funding acquisition:** Bang Nguyen Tran.

**Investigation:** Bang Nguyen Tran, Lauren T. Bennett, Cristina Aponte.

**Methodology:** Bang Nguyen Tran, Mihai A. Tanase, Cristina Aponte.

**Project administration:** Bang Nguyen Tran.

**Resources:** Bang Nguyen Tran, Lauren T. Bennett.

**Software:** Bang Nguyen Tran, Mihai A. Tanase.

**Supervision:** Mihai A. Tanase, Lauren T. Bennett, Cristina Aponte.

**Validation:** Bang Nguyen Tran.

**Visualization:** Bang Nguyen Tran, Lauren T. Bennett, Cristina Aponte.

**Writing – original draft:** Bang Nguyen Tran.

**Writing – review & editing:** Bang Nguyen Tran, Mihai A. Tanase, Lauren T. Bennett, Cristina Aponte.

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
