## [Decision Letter · Decision Letter 0]

24 Jun 2020

PONE-D-20-11188

High-severity wildfires in temperate Australian forests have increased in extent and aggregation in recent decades

PLOS ONE

Dear Dr. TRAN,

Thank you for submitting your manuscript to PLOS ONE. After careful consideration, we feel that it has merit but does not fully meet PLOS ONE’s publication criteria as it currently stands. Therefore, we invite you to submit a revised version of the manuscript that addresses the points raised during the review process.

We look forward to receiving your revised manuscript.

Kind regards,

Krishna Prasad Vadrevu, Ph.D

Academic Editor

PLOS ONE

Additional Editor Comments (if provided):

Dear authors,

Please kindly address the important concerns raised by the reviewers as below:

-Address the novelty and terminology issues

-Clarifications relating to patch shape complexity and inherent image quality issues.

-Explanation regarding the use of fixed thresholds with spectral indices is not ideal for sensitive detection because it does not take into account local conditions (soil type, drought etc).

-More details and justification on the fire severity mapping technique.

Please see the detailed comments below.

Journal Requirements:

2) Please remove your figures from within your manuscript file, leaving only the individual TIFF/EPS image files, uploaded separately.  These will be automatically included in the reviewers’ PDF.

3) Please include captions for your Supporting Information files at the end of your manuscript, and update any in-text citations to match accordingly. Please see our Supporting Information guidelines for more information: http://journals.plos.org/plosone/s/supporting-information.

Reviewers' comments:

Reviewer's Responses to Questions

**Comments to the Author**

1. Is the manuscript technically sound, and do the data support the conclusions?

Reviewer #1: No

Reviewer #2: Yes

Reviewer #3: No

2. Has the statistical analysis been performed appropriately and rigorously? 

Reviewer #1: Yes

Reviewer #2: Yes

Reviewer #3: Yes

3. Have the authors made all data underlying the findings in their manuscript fully available?

Reviewer #1: Yes

Reviewer #2: Yes

Reviewer #3: Yes

4. Is the manuscript presented in an intelligible fashion and written in standard English?

Reviewer #1: Yes

Reviewer #2: Yes

Reviewer #3: Yes

5. Review Comments to the Author

Reviewer #1: This study examines high severity fires in the state of Victoria Australia through a retrospective analysis of wildfires over the last 30 years. The authors conclude that the number of high severity fires has increased over time and speculate that this trend can be of great consequence to the fire-adapted temperate forests of Australia.

While I believe that studies of high severity fires are very important and highly relevant, I cannot see the novelty of this study, specifically, the difference between this study, which the authors claim is ‘for the first time’ (L 287 and L 311), and fire severity maps of [64]? From Table 1 and the Methods it seems that the temporal changes of fire severity have already been mapped on the State level prior to this work and those are of high quality based on in situ and high-resolution image confirmation.

The section on fire severity mapping (L 164) is not clear. Because [64] contains the total and high severity burnt areas (Table 1), and years of fires (the Methods), why authors repeated the analysis using Landsat images? Yet analysed changes in the number of fires and fire size using the data from [64], Fig 2? In the Methods section the authors say they excluded 43 fires and only 162 fires were analysed (L 129), while Fig 2 is based on 211 fires from [64], L 227. What analysis is based on [64] and on images processed in this study?

There is also unclear terminology, e.g. what is the ‘patch’? I understand it reflects high severity burnt areas, but only early in the text it says so (L 86?) and no definition is given when the patch analysis is described.

Please clarify what is the difference between ‘fire area’ and ‘total fire size’ L 209-212? From Fig 2 it looks like unit of ‘fire size’ is ha, and by reading further it seems that the total fire size includes all severity classes, L 222, but it can be only guessed if ‘fire area’ relates to high severity areas? Then how does fire area relate to patch? Is it an aggregation of patches?

Fig 4, because ‘patch’ is not clearly defined, it’s not clear if the analysis relates to the burnt area, high severity area or..?

In the discussion, can it be that the changes in patch shape complexity relate to the image quality as the data from earlier years would come from less sophisticated satellite images such as Landsat 5 rather than indicate increasing severity of fires?

L 329, the reference [86] from California’s conifer forests is not highly relevant to the forest resilience statement regarding fire adapted Eucalyptus of Australian forests. Are there conifer forests in the state of Victoria, subjected to high severity burns?

In conclusion, studies of fire severity changes over time are highly important and relevant yet a clear separation of novelty of this work vs already conducted state level analysis is required.

Reviewer #2: This is a very well written, rigorous and timely paper that leverages the increasing ease of analysing the Landsat satellite archive to examine trends and spatial patterns of severe fire in Victoria, Australia. Fire managers and ecologists are increasingly recognising that fire severity is a vital metric to understand, beyond traditional measures of burnt area. I recommend acceptance of this paper subject to additional comment by the authors on the following issues:

Line 221; official fire history records and databases tend to decline markedly in quality the further back in time one looks; the trend in number of fires per year may therefore to some extent be impacted by how well records of fires were kept in the 1980s - can the authors comment on the quality of the dataset in this regard? Has satellite burnt area mapping confirmed this trend in Victoria independently of the fire history database?

General comments; this study focuses specifically on "wildfires"; no specific mention of prescribed/hazard reduction burns is made so I am assuming they are explicitly excluded. While prescribed fires are intended to be of low severity, and usually are, this is not always the case. Can the authors comment on whether they explicitly excluded prescribed fires, and if so, how can they be sure these excluded fires did not have high severity patches that escaped analysis?

Reviewer #3: Tran and coauthors investigate changes in fire severity (fires >1,000 ha) in south-eastern Australia over the last 30 years. They find that fire severity has increased through time, both in absolute terms as well as in terms of the proportion of area burnt. They also investigate several other properties of fire severity such as patch size, number and clustering, as well as regional variation.

Fire severity provides a clear link between fire and its effects on vegetation and ecosystems more broadly, yet outside of the U.S. there are few studies that have examined long term trends in fire severity. As the authors recognise, given widespread interest, evidence of the existence of trends in fire severity also fills an important gap in our understanding of wildfire and climate change (commentary on the existence of such trends in the absence of evidence notwithstanding).

While I commend the authors for tackling this subject I have serious concerns about the methods they use to measure fire severity. Their use of fixed thresholds with spectral indices is not ideal for sensitive detection because it does not take into account local conditions (soil type, drought etc). As far as I can tell they have not calibrated their severity measurement in this way. This is doubly important because the whole point of looking at changes over time is separating the signal (severity changes) from the noise (eg changes due to climate or based on local differences). Thus there is concern that their method omits key elements of both spatial and temporal variation.

Despite these concerns, the authors’ validation metrics appear reasonable, suggesting their work nevertheless captures some properties of fire severity. However, the performance appears systematically worse than other methods now available (Gibson et al. 2020, Collins et al. 2018, 2020). Thus there are both theoretical and practical reasons for preferring an alternative approach.

A lesser but also important issue is that their fire severity mapping technique is based on work outlined in conference proceedings. Although the proceedings are listed in journal citation databases, I don’t think it is appropriate that a foundational piece of this study comes from there. The method deserves proper scrutiny and I don’t have confidence that the conference proceedings provide that, nor is it reasonable for reviewers to consider this conference proceeding in addition to the manuscript itself.

I have some other more minor comments on the manuscript but do not feel it appropriate to raise them in light of these more substantial issues.

6. PLOS authors have the option to publish the peer review history of their article (what does this mean?). If published, this will include your full peer review and any attached files.

Reviewer #1: No

Reviewer #2: **Yes: **Grant James Williamson

Reviewer #3: No

---

## [Author Response · Author response to Decision Letter 0]

15 Oct 2020

[PONE-D-20-11188] - (Revision 1) - Reply to Reviewers

Title: High-severity wildfires in temperate Australian forests have increased in extent and aggregation in recent decades 

Thank you very much for the invitation to revise our manuscript. We are encouraged that the Reviewers appreciated the work and thank them for their constructive comments that have led to significant improvements in the manuscript.

We have made several changes in response to the Reviewer’s comments as well as several minor changes to further clarify our approach, noting that the paper’s key findings remain unchanged. We trust that the changes as detailed below fully address the Reviewers’ comments, and that the paper will now be accepted for publication.

Please note references to line numbers in the reviewers’ comments refer to our original submission, whereas line numbers in our responses refer to the revised version (with track changes). Underlined text in responses indicates new text.

Reviewer # 1

Our thanks to Reviewer 1 for their insightful comments. We have addressed Reviewer 1’s comments as follows:

1) Comment 1: While I believe that studies of high severity fires are very important and highly relevant, I cannot see the novelty of this study, specifically, the difference between this study, which the authors claim is ‘for the first time’ (L 287 and L 311), and fire severity maps of [64]? From Table 1 and the Methods it seems that the temporal changes of fire severity have already been mapped on the State level prior to this work and those are of high quality based on in situ and high-resolution image confirmation.

Response:

The text detailing the methods was not sufficiently clear, which has led to this confusion: As indicated in the text, the wildfire history dataset available from the Victorian Department of Environment, Land, Water & Planning (‘DELWP’; [64]) contains the spatial extent of the wildfires since 1926. Fire severity mapping in Victoria has not been conducted consistently, with only some fires being assessed for severity in the period between 1998 and 2014. This fire severity information is contained in a spatial layer that we did not cite in the text but that has now been included ([78]) To conduct this study, we had to generate the severity mapping of the selected 162 fires, and we did it by implementing a random forest classification model. The model was trained with the severity data available in the spatial layer provided by the government. Therefore, one of the novelties of this study was the generation of severity maps for all the wildfires larger than 1000 ha that occurred between 1987 and 2017 and for which there were satellite images available, which has indeed been done here for the first time. 

Change: (L170-173) “Severity of the wildfires in Victoria has not been consistently recorded, with historic fire severity mapping only available for nine years in the period between 1998 and 2014[78]. To generate fire severity maps for the 162 selected wildfires ensuring the consistency of the classification we used Fire severity was mapped using a Random “

([78]: Department of Environment, Land, Water & Planning - DELWP. Aggregated Fire Severity Classes from 1998 onward. Melbourne, Victoria, Australia: Department of Environment Land Water and Planning; 2017. Available from https://discover.data.vic.gov.au/dataset/aggregated-fire-severity-classes-from-1998-onward).

Table 1 might have contributed to this confusion as the caption did not clearly indicate that the values were relative to the 162 wildfires, and that the data in the final column were from this study (not the State data). 

Change: (L136-140) To avoid such confusion, we have modified the caption. “Table 1. Characteristics of the bioregions in the study area affected by the selected 162 fires […] Number of wildfires included in this study (i.e. 162 wildfires greater than 1000 ha, occurred between 1987 and 2017 and with available Landsat imagery) and their cumulative total [64] and high-severity burnt area (as estimated in this study).”

2) Comment 2: The section on fire severity mapping (L 164) is not clear. Because [64] contains the total and high severity burnt areas (Table 1), and years of fires (the Methods), why authors repeated the analysis using Landsat images? Yet analysed changes in the number of fires and fire size using the data from [64], Fig 2? In the Methods section the authors say they excluded 43 fires and only 162 fires were analysed (L 129), while Fig 2 is based on 211 fires from [64], L 227. What analysis is based on [64] and on images processed in this study?. 

Response:

As above, we agree that the text was not sufficiently clear. From the fire history dataset, we identified a total of 211 wildfires between 1987 and 2017 that met the criteria of being >1000ha. Those were used to analyze changes in the number and total extent of the fires in the studied period. Of those 211, we generated severity maps for the 162 for which there were pre and post fire Landsat images available and cloud free. Thus, fire severity analysis was conducted only for those 162. 

Change: We have modified the text to clarify when each dataset was used. (L125-134) (Methods): “That amounted to 211 wildfires that were used to assess changes in the number of fires per year and mean fire size between 1987 and 2017. Each fire was classified according to its dominant bioregion [53]. For the purpose of assessing changes in fire severity, 32 of the 211 wildfires were discarded because pre- or post-fire remote sensing images were unavailable, and 11 were discarded because clouds covered more than 25% of the fire affected area, which may affect the spatial metrics assessed in our study. In total, a subset of 162 wildfires, with at least two fires per year over the past three decades, was used to generate fire-severity maps and analyse changes in severity patterns.”

(L239-240) (Results): “Based on the fire history dataset (n=211), the number of wildfires per year larger than 1000 ha between 1987 and 2017 increased significantly (P= 0.012)”

3) Comment 3: There is also unclear terminology, e.g. what is the ‘patch’? I understand it reflects high severity burnt areas, but only early in the text it says so (L 86?) and no definition is given when the patch analysis is described.

Response:

We agree that a clear definition of the term ‘patch’ is missing. The term is used to refer to areas burnt by high-severity fire surrounded by a different severity within the wildfire perimeter. We have added this definition in the description of the spatial configuration metrics. 

Change: (L200-202) ”Spatial configuration metrics were calculated at the patch level, i.e. areas of high-severity fire surrounded by different severities within the wildfire boundary.”

4) Comment 4: Please clarify what is the difference between ‘fire area’ and ‘total fire size’ L 209-212? From Fig 2 it looks like unit of ‘fire size’ is ha, and by reading further it seems that the total fire size includes all severity classes, L 222, but it can be only guessed if ‘fire area’ relates to high severity areas? Then how does fire area relate to patch? Is it an aggregation of patches?

Fig 4, because ‘patch’ is not clearly defined, it’s not clear if the analysis relates to the burnt area, high severity area or..?.

Response:

We agree that the use of both terms ‘fire area’ and ‘total fire size’ is confusing. We have revised the text to consistently use the term ‘fire size’, which has been defined as the total wildfire area (ha). In contrast, we use the qualifier ‘high-severity” to clearly indicate when we refer exclusively to the area burnt at high-severity. 

Changes: (L206-209) (Methods): “Edge density is the ratio between the total length (m) of the edges of the high-severity patches and the fire size (i.e. total wildfire area burnt at any severity; ha)”

(L225-231) (Methods):“Predictor variables included year and fire size (i.e. total wildfire area, ha) as fixed effects and bioregion as a random effect, which was only included in the state-wide mixed effects models. Fire size was included as covariate in all models as it can be”

Figure 4 shows the spatial metrics, which always refer to the patches of high-severity fire. For simplicity and readability, we do not include the qualificative ‘high-severity” in the axes labels. However, we will include a clarification in the caption

Change: (L277-283) “Fig 4. Changes in high-severity spatial metrics over time. Each subplot displays a scatterplot between the Year of the fire and the defined high-severity spatial metric. Dots represent each of the 162 wildfires. Values are the results for single mixed effects models where Year and Fire size are fixed effects and Bioregion is a random effect. Lines represent significant (solid black) or not significant (dashed grey) linear relationships.”

5) Comment 5: In the discussion, can it be that the changes in patch shape complexity relate to the image quality as the data from earlier years would come from less sophisticated satellite images such as Landsat 5 rather than indicate increasing severity of fires?

Response:

The Landsat Program represents the world's longest continuously-acquired collection of space-based moderate-resolution land remote sensing data and thus it provides essential land change data and trending information not otherwise available. The program has been designed to ensure its capability to track changes overtime is preserved. To that end the technical prescriptions (e.g. spectral bands, bandwidths, spatial resolution) of its sensors have remained consistent through the different Landsat missions. That consistency has made the Landsat time-series one of the most widely used to monitor land surface changes overtime. 

In accordance, we have no reason to suspect that the changes observed in the high-severity spatial patterns could be related to the characteristics of the sensors. 

In addition to that, all images were obtained from the Landsat Collection 1 Tier 1, which according to the USGS have the highest available data quality and are considered suitable for time-series analysis. Tier 1 includes Level-1 Precision and Terrain corrected data that have well-characterized radiometry and are inter-calibrated across the different Landsat instruments. The georegistration of Tier 1 scenes is consistent and within prescribed image-to-image tolerances of ≦ 12-meter radial root mean square error (RMSE) (https://www.usgs.gov/core-science-systems/nli/landsat/landsat-collection-1) .

Change: (L145-146) “Wildfire severity of the selected 162 fires was mapped using Landsat TM, ETM+ and Landsat 8 imagery (30 m spatial resolution, all from Landsat Collection 1, Tier 1)”.

6) Comment 6: L 329, the reference [86] from California’s conifer forests is not highly relevant to the forest resilience statement regarding fire adapted Eucalyptus of Australian forests. Are there conifer forests in the state of Victoria, subjected to high severity burns?

Response:

We intended to show that increases in high-severity fire are concerning because of their impact on resilience worldwide but also agree that a reference related to the impact of fire severity on Victorian temperate forest would be relevant. We have modified the text to clarify our message. 

Change: (L356-359): “Our quantified increases in high-severity burned area can lead to concerns about the resilience of Victoria’s temperate forests [20, 93], similar to those expressed for other forest types elsewhere [4, 92, 95].”

7) Comment 7: In conclusion, studies of fire severity changes over time are highly important and relevant yet a clear separation of novelty of this work vs already conducted state level analysis is required.

Response:

As indicated before, state fire history dataset only contained the extent of the wildfires. The generation of fire severity maps and the analysis of the high-severity fire metrics is all an original work developed in this study.

Reviewer # 2

Our thanks to Reviewer 2 for their positive assessment that the paper ‘is a very well written’, ‘it is rigorous and timely paper that leverages the increasing ease of analysing the Landsat satellite archive to examine trends and spatial patterns of severe fire in Victoria, Australia’, and ‘Fire managers and ecologists are increasingly recognising that fire severity is a vital metric to understand, beyond traditional measures of burnt area’. We have addressed Reviewer 2’s comments as follows: 

8) Comment 1: Line 221; official fire history records and databases tend to decline markedly in quality the further back in time one looks; the trend in number of fires per year may therefore to some extent be impacted by how well records of fires were kept in the 1980s - can the authors comment on the quality of the dataset in this regard? Has satellite burnt area mapping confirmed this trend in Victoria independently of the fire history database?

Response:

We thank the referee for the useful comment. We agree that the accuracy of the records has changed over time, something that may be particularly true for small fires that were unaccounted for. However, we believe this may not have impacted the quality of our dataset as we focused on wildfires larger than a 1000ha. Furthermore, the trend was still consistent when reducing the study period to remove the initial years where the records could have been more uncertain. Unfortunately, no satellite burnt area mapping has been conducted to confirm this trend independently. 

9) Comment 2: General comments; this study focuses specifically on "wildfires"; no specific mention of prescribed/hazard reduction burns is made so I am assuming they are explicitly excluded. While prescribed fires are intended to be of low severity, and usually are, this is not always the case. Can the authors comment on whether they explicitly excluded prescribed fires, and if so, how can they be sure these excluded fires did not have high severity patches that escaped analysis?

Response:

Our study focused only on “wildfires” and thus we explicitly excluded prescribed fires from the fire history data available from the Victorian Department of Environment, Land, Water & Planning of Victoria state at the beginning of the wildfire selection. It was out of the scope of this study to investigate any changes in the severity of the prescribed fires, most of which did not meet the minimum 1000ha criteria. 

Reviewer # 3

Our thanks to Reviewer 3 for their positive assessment that the paper’s ‘evidence of the existence of trends in fire severity also fills an important gap in our understanding of wildfire and climate change’ and ‘examined long term trends in fire severity’. We have addressed Reviewer 3’s comments as follows: 

1) Comment 1: Fire severity provides a clear link between fire and its effects on vegetation and ecosystems more broadly, yet outside of the U.S. there are few studies that have examined long term trends in fire severity. As the authors recognise, given widespread interest, evidence of the existence of trends in fire severity also fills an important gap in our understanding of wildfire and climate change (commentary on the existence of such trends in the absence of evidence notwithstanding).

While I commend the authors for tackling this subject I have serious concerns about the methods they use to measure fire severity. Their use of fixed thresholds with spectral indices is not ideal for sensitive detection because it does not take into account local conditions (soil type, drought etc). As far as I can tell they have not calibrated their severity measurement in this way. This is doubly important because the whole point of looking at changes over time is separating the signal (severity changes) from the noise (eg changes due to climate or based on local differences). Thus there is concern that their method omits key elements of both spatial and temporal variation. 

Response:

We apologize for the lack of clarity in the description of the methods implemented on the fire severity mapping. Fire severity classification was not conducted based on fixed threshold with spectral indices, as understood by Reviewer 3. Instead, and as indicated in the ‘Fire severity mapping’ section, we used a Random Forest (RF) classification model trained with 2238 reference plots (60% of the entire reference dataset) from eight large wildfires plots between 1998 and 2009 that covered all forest types encompassed in this study. A total of 12 predictor variables were included in the RF classification model: the identified four best performing spectral indices for the studied forest types (dNBR, dNDVI, dNDWI, dMSAVI) and their pre- and post- fire values; The RF algorithm was validated on an independent set of 1492 reference plots (40% of the reference dataset), yielding a high classification accuracy for the high-severity fire, with a commission error (plots wrongly attributed to high severity) of 0.06 and an omission error (high severity plots incorrectly classified) of 0.18. RF classification models are known to outperform single thresholding and are being increasingly and more widely implemented. 

On the other hand, as explained in the methods section, we limited the influence of spatial variation by using a change detection approach, where severity is classified at the pixel level based on the differences between pre-and post- fire signal. The influence of temporal variation was also reduced by selecting pre- and post-fire images that were not more than 3 months apart, thus minimizing the impact of phenology and atmospheric conditions. 

Change: (L170-182) For clarification, we have added details to the description of the fire severity classification. “Fire severity was mapped using a Random Forest model based on spectral indices that had been previously trained and validated by the authors for the same study area [61]. The reference fire-severity dataset used for training and validation was comprised of 3730 plots from eight large wildfires (>5,000 ha) that occurred between 1998 and 2009 and covered 13 forest types differing in species composition, canopy cover, canopy height and regeneration strategy. These forest types match those affected by the 162 wildfires of this study. Fire severity of the 3730 reference plots had been assessed in situ or visually interpreted on very high resolution orthophotos by the Department of Environment, Water & Planning (DELWP). Severity was classified as […]”

2) Comment 2: Despite these concerns, the authors’ validation metrics appear reasonable, suggesting their work nevertheless captures some properties of fire severity. However, the performance appears systematically worse than other methods now available (Gibson et al. 2020, Collins et al. 2018, 2020). Thus, there are both theoretical and practical reasons for preferring an alternative approach.

Response:

As clarified above, in this study we have used a Random Forest classification model to map fire severity, which is the same method implemented by Gibson et al. 2020, Collins et al. 2018, 2020. We trust that therefore there is no further concerns regarding the severity mapping conducted in this study. 

3) Comment 3: A lesser but also important issue is that their fire severity mapping technique is based on work outlined in conference proceedings. Although the proceedings are listed in journal citation databases, I don’t think it is appropriate that a foundational piece of this study comes from there. The method deserves proper scrutiny and I don’t have confidence that the conference proceedings provide that, nor is it reasonable for reviewers to consider this conference proceeding in addition to the manuscript itself.

Response:

We agree that the methods used to conduct a study should be sufficiently explained in the text of the manuscript so that there is no need for the readers to access further publications. That is why the ‘Fire severity mapping’ section includes a detailed explanation of the development, validation and accuracy of the random forest model used in this study. Further details have been added to the text, thus in our opinion the information now provided is sufficient for the reader to understand the method and gauge the robustness of the approach. 

The conference paper, which was published by SPIE remote sensing after a review process, compared the classification accuracy between Random Forest model and the SI threshold approach, which confirmed that Random Forest models outperform thresholding. 

4) Comment 4: I have some other more minor comments on the manuscript but do not feel it appropriate to raise them in light of these more substantial issues.

Response:

We will be glad to address the minor comments in any subsequent revision

---

## [Decision Letter · Decision Letter 1]

4 Nov 2020

High-severity wildfires in temperate Australian forests have increased in extent and aggregation in recent decades

PONE-D-20-11188R1

Dear Dr. TRAN,

We’re pleased to inform you that your manuscript has been judged scientifically suitable for publication and will be formally accepted for publication once it meets all outstanding technical requirements.

Kind regards,

Krishna Prasad Vadrevu, Ph.D

Academic Editor

PLOS ONE

Additional Editor Comments (optional):

Reviewers' comments:

Reviewer's Responses to Questions

**Comments to the Author**

1. If the authors have adequately addressed your comments raised in a previous round of review and you feel that this manuscript is now acceptable for publication, you may indicate that here to bypass the “Comments to the Author” section, enter your conflict of interest statement in the “Confidential to Editor” section, and submit your "Accept" recommendation.

Reviewer #2: (No Response)

2. Is the manuscript technically sound, and do the data support the conclusions?

Reviewer #2: Yes

3. Has the statistical analysis been performed appropriately and rigorously? 

Reviewer #2: Yes

4. Have the authors made all data underlying the findings in their manuscript fully available?

Reviewer #2: Yes

5. Is the manuscript presented in an intelligible fashion and written in standard English?

Reviewer #2: Yes

6. Review Comments to the Author

Reviewer #2: (No Response)

7. PLOS authors have the option to publish the peer review history of their article (what does this mean?). If published, this will include your full peer review and any attached files.

Reviewer #2: **Yes: **Grant James Williamson

---

## [Editor Report · Acceptance letter]

9 Nov 2020

PONE-D-20-11188R1 

High-severity wildfires in temperate Australian forests have increased in extent and aggregation in recent decades 

Dear Dr. TRAN:

I'm pleased to inform you that your manuscript has been deemed suitable for publication in PLOS ONE. Congratulations! Your manuscript is now with our production department. 

Kind regards, 

on behalf of

Dr Krishna Prasad Vadrevu 

Academic Editor

PLOS ONE